# Role of B-Cell Translocation Gene 1 in the Pathogenesis of Endometriosis

**DOI:** 10.3390/ijms20133372

**Published:** 2019-07-09

**Authors:** Jeong Sook Kim, Young Sik Choi, Ji Hyun Park, Jisun Yun, Soohyun Kim, Jae Hoon Lee, Bo Hyon Yun, Joo Hyun Park, Seok Kyo Seo, SiHyun Cho, Hyun-Soo Kim, Byung Seok Lee

**Affiliations:** 1Department of Obstetrics and Gynecology, University of Ulsan College of Medicine, Ulsan University Hospital, Ulsan 44033, Korea; 2Department of Obstetrics and Gynecology, Severance Hospital, Yonsei University College of Medicine, Seoul 03722, Korea; 3Institute of Women’s Life Medical Science, Yonsei University College of Medicine, Seoul 03722, Korea; 4Department of Obstetrics and Gynecology, Gangnam Severance Hospital, Yonsei University College of Medicine, Seoul 06273, Korea; 5Department of Pathology and Translational Genomics, Samsung Medical Center, Sungkyunkwan University School of Medicine, Seoul 06351, Korea

**Keywords:** BTG1, endometriosis, human endometrial stromal cells

## Abstract

Estrogen affects endometrial cellular proliferation by regulating the expression of the *c-myc* gene. B-cell translocation gene 1 (BTG1), a translocation partner of the *c-myc*, is a tumor suppressor gene that promotes apoptosis and negatively regulates cellular proliferation and cell-to-cell adhesion. The aim of this study was to determine the role of BTG1 in the pathogenesis of endometriosis. *BTG1* mRNA and protein expression was evaluated in eutopic and ectopic endometrium of 30 patients with endometriosis (endometriosis group), and in eutopic endometrium of 22 patients without endometriosis (control group). The effect of BTG1 downregulation on cellular migration, proliferation, and apoptosis was evaluated using transfection of primarily cultured human endometrial stromal cells (HESCs) with *BTG1* siRNA. *BTG1* mRNA expression level of eutopic and ectopic endometrium of endometriosis group were significantly lower than that of the eutopic endometrium of the control group. Migration and wound healing assays revealed that BTG1 downregulation resulted in a significant increase in migration potential of HESCs, characterized by increased expression of matrix metalloproteinase 2 (MMP2) and MMP9. Downregulation of BTG1 in HESCs significantly reduced Caspase 3 expression, indicating a decrease in apoptotic potential. In conclusion, our data suggest that downregulation of BTG1 plays an important role in the pathogenesis of endometriosis.

## 1. Introduction

Endometriosis is a commonly occurring gynecological disorder that may cause pelvic pain and infertility [1]. The diagnosis of endometriosis is pathologically confirmed by the ectopic presence of endometrial-like epithelium and stroma. Endometriosis is prevalent in approximately 10% of all reproductive women, 20–30% of women with infertility, and 40–80% of women with chronic pelvic pain, or both [2]. Although several pathogenic theories have been suggested, the exact underlying mechanisms of endometriosis still remains unclear. The long-term management of endometriosis-related symptoms and recurrence after surgery also remains a challenge. Understanding endometrial cellular expression of certain genes and proteins may result in improved diagnosis and treatment of endometriosis.

Cellular proliferation, differentiation, and apoptosis are cell cycle-dependent processes [3]. Disruption of cell cycle regulation results in tumor formation and progression [4]. Estrogen regulates endometrial cellular proliferation by binding to an atypical *cis*-element in *c-myc* promoter [5]. Serum estrogen levels are 5–8 times higher than normal in women with endometriosis [6]. Estrogen receptor-α and *c-myc* are highly expressed in deeply infiltrating endometriotic tissue [7]. Therefore, the *c-myc* gene may play a crucial role in endometriosis.

B-cell translocation gene 1 (*BTG1*) was identified as a translocation partner of the *c-myc* gene in B-cell chronic lymphocytic leukemia [8]. BTG1 belongs to the BTG/TOB family of anti-proliferative proteins that include BTG2, BTG3, TOB1, and TOB2, which regulates cell growth and differentiation [9,10,11]. BTG1 is primarily expressed in quiescent cells at G_0_/G_1_ phase of cell cycle, and its level declines as cells enter S phase [12]. Exogenous overexpression of BTG1 reduces proliferation of murine fibroblasts by inducing G1 arrest and/or apoptosis [12]. BTG1 also negatively regulate proliferation of murine microglial cells [13].

*BTG1* is also a known tumor suppressor gene that negatively regulates cellular proliferation, cell-to-cell adhesion, migration, and invasion in carcinomas of esophagus, nasopharynx, liver, breast, kidneys and lungs [14,15,16,17,18]. We recently demonstrated that BTG1 expression is significantly reduced in ovarian carcinoma tissues, and that BTG1 silencing in ovarian carcinoma occurs through epigenetic repression [19].

The role of BTG1 in endometrial cellular proliferation and the precise molecular mechanisms of BTG1 function in the pathogenesis of endometriosis remain unclear. In this study, we investigated BTG1 expression in eutopic and ectopic endometrial tissue samples obtained from patients with or without endometriosis. We also analyzed the expression levels of *BTG1* mRNA and protein and several cell cycle-related genes in primarily culture human endometrial stromal cells (HESCs), and examined whether downregulation of BTG1 affects cellular migration, proliferation, and/or apoptosis of HESCs.

## 2. Results

### 2.1. Clinical Characteristics

The clinical characteristics of the participants are shown in Table 1. The endometriosis group showed significantly decreased gravidity (*p* = 0.034) and parity (*p* = 0.006) compared to the control group. Patients with endometriosis exhibited lower body mass index than those without endometriosis, even though the difference between the groups was not statistically significant (*p* = 0.092). Serum levels of cancer antigen 125 were significantly higher in the endometriosis group than in the control group (*p* = 0.002). The differences in visual analogue scale (VAS) for pain between the groups were statistically significant. Mean VAS were significantly higher in the endometriosis group (6.57 ± 8.43) than in the control group (1.55 ± 9.88; *p* < 0.001).

### 2.2. Expression of BTG1 mRNA and Protein in Eutopic and Ectopic Endometrium of Patients with and without Endometriosis

Eutopic endometrium of the control group (1.95 ± 0.35) displayed significantly higher *BTG1* mRNA expression than both eutopic (1.12 ± 0.14; *p* = 0.048) and ectopic (0.81 ± 0.22; *p* = 0.004) endometrium of the endometriosis group (Figure 1A). In both groups, differences in *BTG1* mRNA expression between proliferative and secretory phases were not statistically significant, although the endometriosis groups exhibited lower expression levels than the control group in both the proliferative and secretory endometrial tissue samples (Table 2).

Using immunohistochemistry, we investigated BTG1 protein expression in eutopic and ectopic endometrial tissue samples in patients with and without endometriosis. Representative photomicrographs showing BTG1 immunoreactivity are shown in Figure 1B. Eutopic endometrial tissue of the control group showed strong BTG1 immunoreactivity in the endometrial glands and stroma. BTG1 was localized in both the nucleus and the cytoplasm of glandular epithelial and stromal cells. In contrast, BTG1 immunostaining was patchy and markedly reduced in the eutopic endometrium of the endometriosis group. About half of the glandular epithelial and stromal cells showed mild-to-moderate staining intensity. BTG1 staining intensity in eutopic and ectopic endometrium of the endometriosis group was markedly decreased compared with that in the eutopic endometrial tissue samples of the control group (Figure 1C). BTG1 was almost absent in the endometrial-type glands and stroma, although a faint nonspecific cytoplasmic staining was observed in the glandular epithelium. The BTG1 immunostaining results were in agreement with the mRNA expression data.

### 2.3. Effect of BTG1 Downregulation on Migration Potential of HESCs

We observed that the expression levels of BTG1 was reduced in endometriosis. Therefore, we speculated that downregulation of BTG1 may have effects on the migration of HESCs. We confirmed a reduction in BTG1 expression after *BTG1* siRNA transfection of HESCs (Figure 2A), and then examined the mRNA expression levels of *MMP2* and *MMP9*, which encode matrix metalloproteinase 2 (MMP2) and MMP9, respectively. They are well-known indicators for cell migration [20,21,22,23], and previous studies have shown that expression of MMP2 and/or MMP9 were elevated in endometriosis [24,25,26]. RT-PCR revealed that *BTG1* siRNA-transfected cells exhibited 1.18- and 1,94-fold increases in *MMP2* and *MMP9* mRNA expression levels compared to vehicle-treated control cells, with marginal significance (*p* = 0.059 and *p* = 0.089, respectively; Figure 2B). In line with this finding, the expression levels of MMP2 and MMP9 proteins were significantly increased in the transfected cells compared with the control cells (Figure 2C), indicating that BTG1 downregulation in HESCs induces an increase in their migration potential.

We further performed migration (Figure 2D) and wound healing (Figure 2E) assays. Compared to the control cells (4.40 ± 0.49), a significantly higher number of migrating cells was observed in the transfected cells (41.80 ± 1.07; *p* < 0.001). Wound healing assay revealed that scratched areas filled up with more HESCs in the transfected cells compared with the control cells (*p* = 0.005). Taken together, our observations suggest that BTG1 suppresses migration potential of in HESC.

### 2.4. Effect of BTG1 Downregulation on Apoptotic Potential and Proliferative Activity of HESCs

To investigate the role of BTG1 in apoptosis of HESCs, the mRNA expression levels of *Caspase 3*, *Caspase 8*, *Fas*, and *FasL* were measured in HESCs transfected with *BTG1* siRNA or treated with vehicle control (Figure 3A). Among them, *Caspase 3* mRNA expression was significantly affected by BTG1 downregulation. The transfected cells showed a 0.78-fold decrease in the expression level of *Caspase 3* mRNA compared to the control cells (*p* = 0.029). To validate this finding, we evaluated the expression levels of pro- and anti-apoptotic proteins including Caspase 3, cleaved Caspase 3, Bax, and Bcl-2 (Figure 3B). BTG1 downregulation reduced the expression levels of pro-apoptotic factors (cleaved Caspase 3 and Bcl-2) and elevated the expression level of anti-apoptotic factor (Bax). *Fas* mRNA and protein expression appeared to be increased in the transfected cells, the alterations were not statistically significant. Flow cytometry analysis (Figure 4A) revealed that the percentage of Annexin V-positive cells decreased significantly after transfection (6.34 ± 0.34 versus 4.69 ± 0.07%; *p* = 0.008). In addition, MTT assay (Figure 4B) revealed that relative cell proliferation of HESCs was increased significantly after BTG1 downregulation. These findings indicate that downregulation of BTG1 in HESCs induced a decrease in apoptotic potential and an increase in cellular proliferation.

## 3. Discussion

Although *BTG1* is a known anti-proliferative gene, the role of BTG1 in the pathophysiology of endometriosis has remained unclear. In this study, we observed that BTG1 expression was significantly reduced in ectopic and eutopic endometrial tissues of patients with endometriosis. Downregulation of BTG1 increased proliferative activity and migration potential of HESCs, and decreased their apoptotic potential. Our observations suggest that reduced expression of BTG1 facilitates proliferation and migration of HESCs and suppresses their apoptosis, resulting in the progression of endometriosis.

MMPs are enzymes that degrade extracellular matrix proteins and are important for the tissue remodeling process [27]. MMP2 (gelatinase A) and MMP9 (gelatinase B) degrade type IV collagen and fibronectin [20]. Previous studies reported that MMP2 expression is increased along with other proteases in the late secretory endometrium [23], and that the expression level of MMP9 is elevated in patients with endometriosis, suggesting that their endometrial tissue is inherently more invasive [22]. In this study, we demonstrated that downregulation of BTG1 in HESCs resulted in elevated MMP2 and MMP9 expression levels, suggesting that BTG1 negatively regulates endometrial cell migration.

The apoptotic signaling is classified into two major pathways: extrinsic or cytoplasmic and intrinsic pathways. The extrinsic pathway is regulated by the Fas death receptor, and the intrinsic pathway is regulated by the Bcl-2 family of proteins [28,29]. The ratio of anti-apoptotic Bcl-2 to pro-apoptotic Bax is a critical factor for the intrinsic pathway. Hetero-dimerization of Bcl-2 with Bax, which inhibits apoptosis, is controlled by a family of cysteinyl proteases called caspases [30,31,32]. In this study, we showed that cellular apoptosis was reduced after downregulation of BTG1 in HESCs. The expression levels of *Caspase 3* mRNA were significantly reduced after BTG1 downregulation, whereas there were no significant alterations in mRNA expression levels of *Caspase 8*, *Fas*, or *FasL*. The *BTG1* siRNA-transfected HESCs also displayed decreased Bax expression and increased Bcl-2 expression. A previous study documented that overexpression of BTG1 induces apoptosis of breast carcinoma cells, accompanied by a decline in the Bcl-2 level and an increase in the Bax and Caspase 3 levels [18]. Our data suggest that BTG1 regulates cellular apoptosis via the intrinsic pathway in HESCs, thus affecting the pathogenesis of endometriosis.

There are a few limitations in this study. First, we conducted a pilot study using a relatively small number of the endometriosis subjects. Nevertheless, a significant reduction in BTG1 expression was observed in ectopic and eutopic endometrium of patients with endometriosis. Second, the control group did not include disease-free subjects. Since surgical resection of nonpathological ovarian or endometrial tissues from healthy patients raises ethical implications, we inevitably used tissue samples obtained from patients with ovarian mature teratoma, ovarian serous cystadenoma, or paratubal cyst. These benign ovarian lesions are not associated with ovarian endometriosis or endometrial pathology. Third, the majority of patients had moderate-to-severe endometriosis; therefore, we failed to investigate the association between BTG1 expression status and the severity of endometriosis. Fourth, we did not observe any expression of BTG1 in the Ishikawa cell line before and after BTG1 downregulation. We additionally measured MMP2 and MMP9 expression levels in Ishikawa cell line, but there was no difference in the expression levels (Figure 5). Since BTG1 is a tumor suppressor and the Ishikawa cell line is derived from endometrial carcinoma, lack of BTG1 expression was an expected finding.

In conclusion, we demonstrated that downregulation of BTG1 could play an important role in the progression of endometriosis. We observed that BTG1 expression was significantly reduced in ectopic and eutopic endometrial tissues of patients with endometriosis. BTG1 downregulation via siRNA transfection increased migration potential and proliferative activity of HESCs, and decreased their apoptotic potential.

## 4. Materials and Methods

### 4.1. Patients and Tissue Samples

This study (3-2015-0250, 23 November 2015) was reviewed and approved by the Institutional Review Board of Gangnam Severance Hospital (Seoul, Korea). Between 1 June 2015 and 31 July 2015, 54 women who underwent laparoscopy for various gynecological conditions (pelvic mass, pelvic pain, endometriosis, and infertility) were enrolled after obtaining their written informed consent. Thirty-two and 22 patients participated in the endometriosis and control groups, respectively. Their ages ranged from 19 to 45 years. Patients with postmenopausal symptoms, previous hormone or gonadotropin-releasing hormone agonist use, uterine adenomyosis, endometrial lesion (polyp, hyperplasia, or malignancy), infectious disease, acute or chronic inflammatory disease, autoimmune disease, and cardiovascular disease were excluded.

All probable endometriotic lesions were surgically resected and sent to the pathology department for histopathological examination. Patients were assigned to the endometriosis group only after pathological confirmation of endometriosis. The extent of endometriosis was determined according to the revised American Society for Reproductive Medicine Classification of endometriosis [33]. Among 32 patients with peritoneal and/or ovarian endometriosis, 30 patients exhibited a moderate-to-severe form of endometriosis, while the remaining 2 patients had a mild form of the disease. The presence and intensity of endometriosis-related pain, including dysmenorrhea, deep dyspareunia, and/or non-menstrual pelvic pain, were assessed using VAS [34].

Fifteen of the 22 patients who participated in the control group were diagnosed as having mature cystic teratoma, and the remaining 7 patients had serous cystadenoma (5 patients) or paratubal cyst (2 patients). These benign ovarian lesions were not associated with ovarian endometriosis or endometrial pathology. Since surgical resection of nonpathological ovarian or endometrial tissues from healthy patients raises ethical implications, the control group did not include disease-free subjects. Endometrial tissue samples were obtained using a Pipelle catheter.

### 4.2. Culture of Primary Endometrial Stromal Cells and Ishikawa Cell Line

We cultured primary endothelial stromal cells as previously described [35]. Endometrial tissue was finely minced and the cells were dispersed by incubation in Hanks Balanced Salt Solution containing 4-(2-hydroxyethyl)-1-piperazineethanesulfonic acid (2 mmol/mL), 1% penicillin/streptomycin, and collagenase (1 mg/mL, 15 U/mg) for 60 min at 37 °C with agitation and pipetting. The cells were centrifuged and cell pellets were washed, suspended in Dulbecco’s modified Eagle’s medium:Ham F12 (1:Z1) solution containing 10% fetal bovine serum and 1% penicillin/streptomycin, passed through a 40-µm cell strainer (Corning Inc., Corning, NY, USA), and plated on commercially available 75 cm^2^ tissue culture Falcon flasks (BD Biosciences, San Jose, CA, USA). We used cultured HESCs at 3–5 passages for analysis. We cultured the Ishikawa cell line in Minimum Essential Media (Invitrogen, Carlsbad, CA, USA) containing 2.0 mmol/L 1-glutamine and Earl salts, supplemented with 10% fetal bovine serum, 1% sodium pyruvate, and 1% penicillin/streptomycin.

### 4.3. Cell Transfection

Cells were seeded in 6-well plates, cultured to 70–80% confluence, and transfected with *BTG1* siRNA or control siRNA-A (Santa Cruz, Dallas, TX, USA) using Lipofectamine 3000 (Invitrogen) according to the manufacturer’s recommendations at a final concentration of 50 nM. The transfected cells were harvested after 48 h.

### 4.4. RNA Isolation and Quantitative Real-Time Polymerase Chain Reaction (qRT-PCR)

To measure *BTG1* mRNA levels, total RNAs were isolated from cultured cells using the RNeasy Mini Kit (Qiagen Inc., Valencia, CA, USA). RNA sample concentrations were analyzed using a Nanodrop 2000 spectrophotometer (Thermo Fisher Scientific, Waltham, MA, USA). The Superscript III kit (Invitrogen) was used to synthesize cDNA using 1 µg of total RNA primed with oligo(dT). PCR was performed in a C1000 Thermal Cycler (Bio-Rad Laboratories, Hercules, CA, USA). The synthesized cDNA products were stored at −20 °C. qPCR was performed with 2 µL of synthesized cDNA as template using a 7300 Real-Time PCR System (Applied Biosystems, Foster City, CA, USA). Real-time PCR was performed using the Power SYBR Green PCR master mix (Applied Biosystems). The reaction mixture included the cDNA template, forward and reverse primers, ribonuclease-free water, and the SYBR Green PCR master mix at a final reaction volume of 20 µL. Reactions were performed at 95 °C for 5 min, followed by 40 cycles of 95 °C for 30 s, 60 °C for 30 s, 72 °C for 1 min, and a final extension at 72 °C for 5 min. Threshold cycle (*C*t) values and melting curves were calculated using the 7300 Real-time PCR system software (Applied Biosystems). Each reaction was performed in triplicates. If not specified, the mRNA levels in each sample were normalized to those of *GAPDH*. Primer sequences for *BTG1*, *Caspase 3*, *Caspase 8*, *Fas*, *FasL*, *MMP2*, *MMP9* and *GAPDH* are shown in Table 3. The product amount was calibrated to the internal control reference using the ΔΔ*C*t analysis [36].

### 4.5. Immunohistochemistry

BTG1 protein expression was assessed by immunohistochemistry using the Bond Polymer Intense Detection System (Leica Biosystems, Newcastle upon Tyne, UK) according to the manufacturer’s recommendations. Surgically resected tissues were fixed in 10% neutral buffered formalin for 12–24 h. The tissues were then sectioned, processed with an automatic tissue processor, and embedded in paraffin blocks. A rotary microtome was used to cut 4-μm thick sections from each formalin-fixed, paraffin-embedded tissue block. Deparaffinization was performed using Bond Dewax Solution (Leica Biosystems). Antigen retrieval was performed using Bond Epitope Retrieval Solution (Leica Biosystems) for 30 min at 100 °C. Endogenous peroxidases were quenched with hydrogen peroxide for 5 min. Sections were incubated with anti-rabbit antibody against BTG1 (1:100, polyclonal, Abcam, Cambridge, MA, USA) for 15 min at ambient temperature. Biotin-free polymeric horseradish peroxidase-linker antibody conjugate system and Bond-maX automatic slide stainer (Leica Biosystems) were used. Visualization was performed using 1 mM 3,3′-diaminobenzidine, 50 mM Tris-hydrogen chloride buffer (pH 7.6), and 0.006% hydrogen peroxide. The sections were counterstained with hematoxylin. Positive and negative control samples were included with each reaction to minimize inter-assay variation. Normal colonic tissue was used as positive control. The negative control was prepared by replacing the primary antibody with non-immune serum; no detectable staining was evident. The staining intensity of BTG1 was assessed on a scale of 0–3, with 0 indicating negative staining; 1, weak; 2, moderate; and 3, strong.

### 4.6. Protein Extraction and Western Blot Analysis

We prepared protein extracts using a radio-immunoprecipitation assay buffer containing protease and a phosphatase inhibitor cocktail (Thermo Fisher Scientific). We determined the concentrations of total cell lysates using Pierce BCA protein assay kit (Thermo Fisher Scientific). We mixed 20 μg of total protein with 5× sample buffer, and then heated at 95 °C for 5 min. Samples were loaded on 12% sodium dodecyl sulfate-polyacrylamide gels and transferred to a polyvinylidene fluoride membrane (EMD Millipore, Burlington, MA, USA) with use of a Transblot apparatus (Bio-Rad Laboratories). Membranes (EMD Millipore) were blocked with 5% non-fat skim milk in Tris-buffered saline solution (10 mmol/L Tris-hydrogen chloride (pH 7.4) and 0.5 mol/L sodium chloride) with 0.1% *v*/*v* of Tween-20 (TBS-T). The blots were probed with primary antibodies against BTG1 (1:200, Abcam), MMP9 (1:250, Santa Cruz) with TBS-T; MMP2 (1:250, Santa Cruz) with TBS-T, Caspase 3 (1:1,000, Cell Signaling Technology, Danvers, MA, USA) with TBS plus 5% skim milk, Bax (1:500, Santa Cruz) with TBS plus 5% skim milk, Bcl-2 (1:1000, Cell Signaling Technology) with TBS plus 5% skim milk, and GAPDH (1:1000, Santa Cruz) with TBS plus 5% skim milk. They were then incubated in horseradish peroxidase-conjugated secondary antibodies (1:2000, Thermo Fisher Scientific). Proteins were detected using enhanced chemiluminescence (Santa Cruz). The experiment was repeated three times and the data shown are representative.

### 4.7. Migration and Wound Healing Assay

We performed migration assay for BTG1-transfected cells using 8-mm-pore polycarbonate membranes (EMD Millipore) within 24-well plates. Freshly treated by trypsin and washed cells were suspended in 200 mL of serum-free medium and positioned in the top chamber of each insert (5 × 10^4^ cells/well); 600 mL of medium containing 10% fetal bovine serum was added into the lower chambers. Cells were fixed and stained with hematoxylin after incubation for 24 h at 37 °C in a 5% carbon dioxide humidified incubator. Cells in the inner chamber were removed using a cotton swab and the cells that were attached to the bottom side of the membrane were counted and imaged using an inverted microscope (Olympus, Tokyo, Japan) at 200× magnification over ten random fields in each well. For the wound healing assay, HESCs were transfected with control or *BTG1* siRNA for 48 h. The transfected cells were seeded using Dulbecco’s modified Eagle’s medium/F12 (1:1) with 10% fetal bovine serum and antibiotics and cultured in 24-well plates incubated at 37 °C with 5% carbon dioxide for 24 h. A linear wound (scratch) was generated using a sterile 100 µL pipette tip and debris was washed twice with phosphate-buffered saline. Cells were grown in culture for 18–24 h at 37 °C with 5% carbon dioxide. The scratched areas in each well were imaged using the EVOS inverted microscope (Advanced Microscopy Group, Mill Creek, WA, USA) to calculate the migration ability of *BTG1*-knockdown cells using ImageJ Version 1.51, a Java-based image processing program developed at the National Institutes of Health (Bethesda, MD, USA) [37]. The experiments were repeated four times.

### 4.8. Flow Cytometry

Cells were seeded in 6-well plates, incubated with indicated concentrations of chemical for 24 or 48 h, and transfected for 48 h. Cells were washed twice with phosphate-buffered saline; and suspended in 400 mL of binding buffer. Cells were stained with 5 mL Annexin V-fluorescein isothiocyanate (FITC) and 4 mL propidium iodide and incubated at room temperature for 10 min using Annexin V-FITC Apoptosis Detection Kit (BD Biosciences). The stained cells were quantified by flow cytometry using the FACS Canto II system (BD Biosciences) and the data were analyzed using DIVA software (version 8.0; BD Biosciences).

### 4.9. MTT Assay

Cytotoxicity was performed using the CellTiter 96 Non-Radioactive Cell Proliferation Assay (MTT) kit as manufacturer’s recommendations (Promega, Madison, WI, USA). Briefly, 1 × 10^4^ HESCs/well were seeded onto 96 well plates and incubated for 24 hr. After removal of supernatant, the plates were incubated with mixed solution (new media 100 μL and MTT solution 10 μL) for 4 h at 37 °C in a carbon dioxide incubator. Solubilization and stop solution mixture was added. The dark blue formazan product was quantified using a microplate reader at 570 nm (with a 690 nm reference filter; Molecular Device, Sunnyvale, CA, USA).

### 4.10. Statistical Analysis

All data were assessed for normal distribution using the Kolmogorov-Smirnov test or the Shapiro-Wilk test. The Student *t*-test or the Mann-Whitney U test was used for comparisons. One-way analysis of variance was performed in conjunction with the Tukey’s post-hoc test to evaluate differences among groups. Statistical analyses were performed using SPSS for Windows, Version 16.0 (SPSS Inc., Chicago, IL, USA).

## Figures and Tables

**Figure 1 ijms-20-03372-f001:**
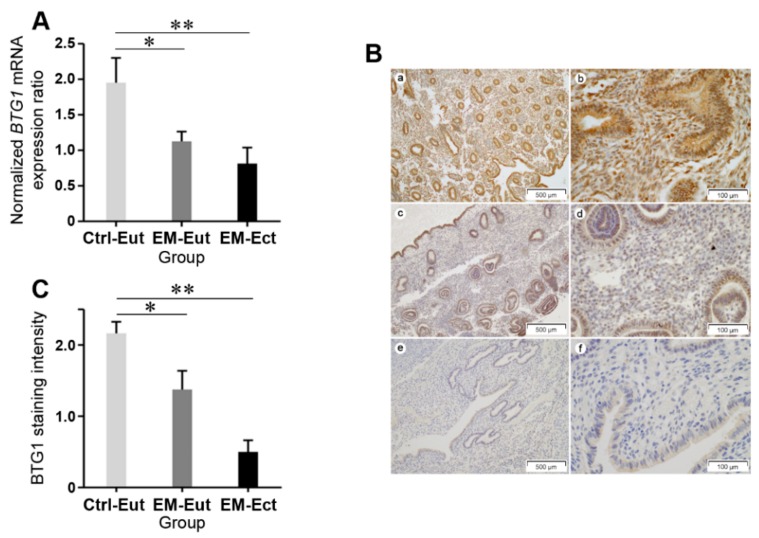
*BTG1* mRNA expression in eutopic and ectopic endometrial tissue samples obtained from patients with or without endometriosis. (**A**) Differences in *BTG1* mRNA expression levels among 3 groups: eutopic endometrium of the control group (Ctrl-Eut) and the endometriosis group (EM-Eut) and ectopic endometrium (EM-Ect). The expression levels of *BTG1* mRNA are normalized to that of *GAPDH* mRNA. (**B**) BTG1 protein expression in eutopic and ectopic endometrial tissue samples. (a) Eutopic endometrial tissue shows uniform and strong BTG1 immunoreactivity in the glandular and stromal cells (×100). (b) High-power view of image a. The cells in the endometrial glands and stroma exhibit strong BTG1 expression both in the nucleus and cytoplasm (×400). (c) Eutopic endometrial tissue of a patient with endometriosis displays patchy BTG1 expression with variable staining intensity (×100). (d) High-power view of image c. About half of the endometrial glandular epithelium and stromal cells show weak-to-moderate nuclear and cytoplasmic BTG1 immunoreactivity (×400). (e) Ectopic endometrial tissue in a patient with endometriosis (×100). (f) High-power view of image e. BTG1 expression is absent in the ectopic endometrial tissue. A faint nonspecific cytoplasmic staining is observed in the glandular epithelium (×400). (**C**) Differences in BTG1 staining intensity among 3 groups. * *p* < 0.05, ** *p* < 0.01. Error bars represent standard error of mean.

**Figure 2 ijms-20-03372-f002:**
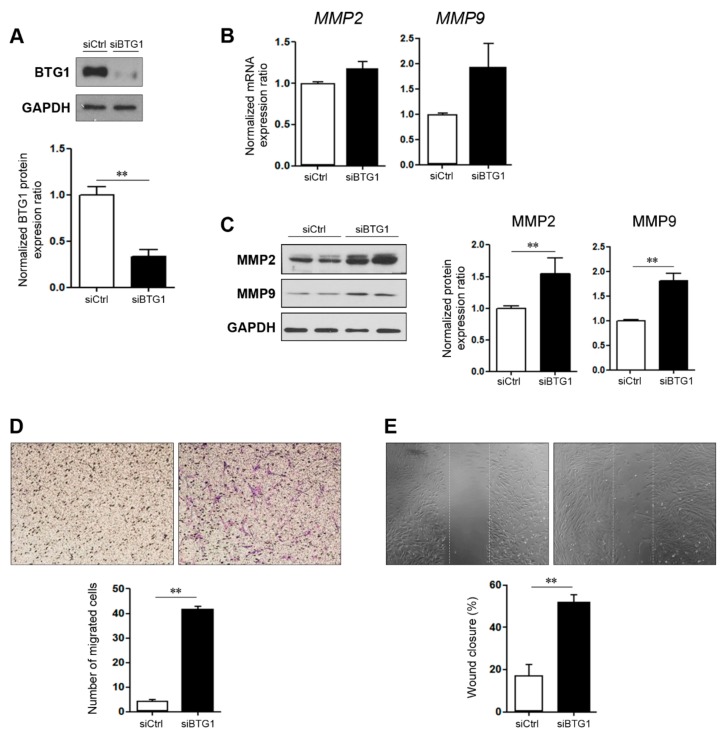
Effect of BTG1 downregulation on migration potential in HESCs. (**A**) Marked reduction in BTG1 protein expression after *BTG1* siRNA transfection. (**B**) The increases in the expression levels of *MMP2* and *MMP9* mRNA after BTG1 downregulation. (**C**,**D**) The significant increases in (C) the expression levels of MMP2 and MMP9 protein and (D) the number of migrated HESCs after BTG1 downregulation (×100). (**E**) Significantly higher rate of wound closure in the *BTG1* siRNA-transfected cells (×100). ** *p* < 0.01. Error bars represent standard error of mean.

**Figure 3 ijms-20-03372-f003:**
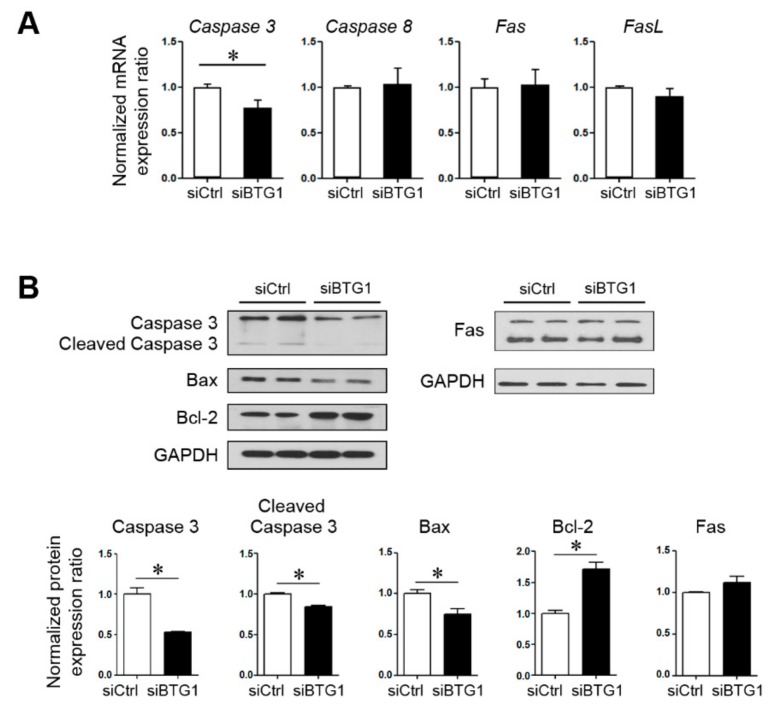
Effect of BTG1 downregulation on the expression of apoptosis-related proteins in HESCs. (**A**) mRNA expression levels of *Caspase 3*, *Caspase 8*, *Fas*, and *FasL* after *BTG1* siRNA transfection. (**B**) Protein expression levels of Caspase 3, cleaved Caspase 3, Bax, Bcl-2, and Fas. * *p* < 0.05. Error bars represent standard error of mean.

**Figure 4 ijms-20-03372-f004:**
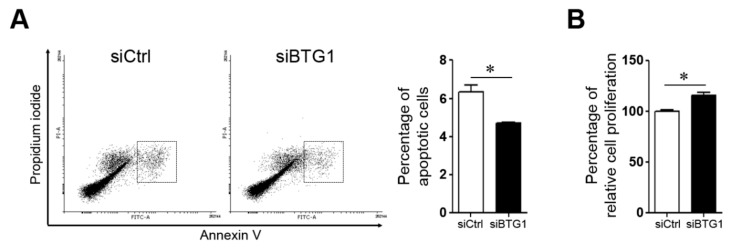
Effect of BTG1 downregulation on apoptotic potential and proliferative activity of HESCs. (**A**) Flow cytometry analysis showing decreased apoptotic potential of HESCs after BTG1 downregulation. A significant decrease in the percentage of apoptosis cells is observed in the *BTG1* siRNA-transfected cells. (**B**) MTT assay showing a significant increase in the percentage of relative cell proliferation after BTG1 downregulation. * *p* < 0.05. Error bars represent standard error of mean.

**Figure 5 ijms-20-03372-f005:**
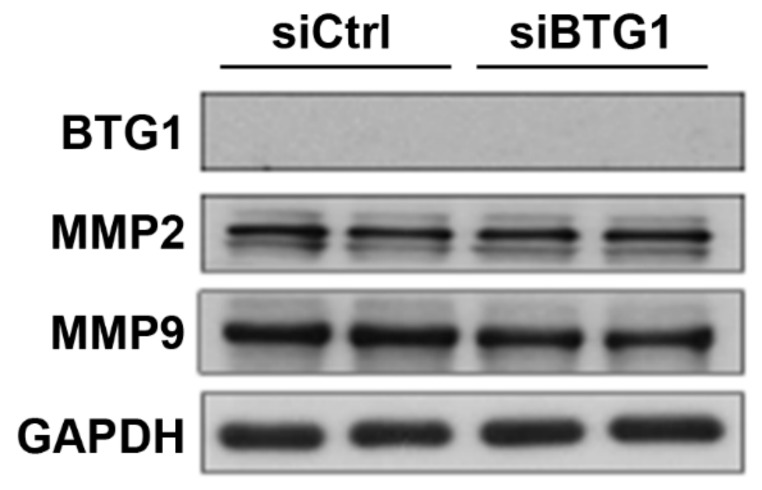
Effect of BTG1 downregulation on the protein expression levels of BTG1, MMP2 and MMP9 in Ishikawa cell line.

**Table 1 ijms-20-03372-t001:** Patient characteristics.

Characteristic	Endometriosis Group	Control Group	*p*-Value
Number of patients	30	22	
Age (years)	34.93 ± 1.33	37.41 ± 1.78	0.349
Gravidity (frequency)	0.90 ± 0.24	1.91 ± 0.38	0.034 *
Parity (frequency)	0.47 ± 0.16	1.23 ± 0.22	0.006 *
Body mass index (kg/m^2^)	20.67 ± 0.34	21.76 ± 0.53	0.092
Cancer antigen 125 (U/mL)	84.06 ± 25.83	18.60 ± 3.17	0.002 *
Visual analogue scale	6.57 ± 0.84	1.55 ± 0.98	<0.001 *

* Statistically significant. Data are expressed as mean ± standard error of mean.

**Table 2 ijms-20-03372-t002:** *BTG1* mRNA expression in endometrium according to the phase of endometrium.

Phase of Endometrium	Group		*p*-Value
	**Endometriosis**	**Control**	
Proliferative phase	1.23 ± 0.21 (*n* = 14)	1.75 ± 0.50 (*n* = 8)	0.570
Secretory phase	1.03 ± 0.20 (*n* = 15)	1.88 ± 0.60 (*n* = 10)	0.091

Data are expressed as mean ± standard error of mean.

**Table 3 ijms-20-03372-t003:** Primer sequences used.

Gene	Sequence	
*BTG1*	Forward	CAA GGG ATC GGG TTA CCG TTG T
	Reverse	AGC CAT CCT CTC CAA TTC TGT AGG
*Caspase 3*	Forward	GGA AGC GAA TCA ATG GAC TCT GG
	Reverse	GCA TCG ACA TCT GTA CCA GAC C
*Caspase 8*	Forward	CCA GAG ACT CCA GGA AAA GAG A
	Reverse	GAT AGA GCA TGA CCC TGT AGG C
*Fas*	Forward	AGC TTG GTC TAG AGT GAA AA
	Reverse	GAG GCA GAA TCA TGA GAT AT
*FasL*	Forward	CAG CTC TTC CAC CTG CAG AAG G
	Reverse	AGA TTC CTC AAA ATT GAT CAG AGA GAG
*MMP2*	Forward	ACC GCG ACA AGA AGT ATG GC
	Reverse	CCA CTT GCG GTC ATC ATC GT
*MMP9*	Forward	CGA TGA CGA GTT GTG GTC CC
	Reverse	TCG TAG TTG GCC GTG GTA CT
*GAPDH*	Forward	ACC ACA GTC CAT GCC ATC AC
	Reverse	TCC ACC ACC CTG TTG CTG TA

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
