# Peer review of "Role of B-Cell Translocation Gene 1 in the Pathogenesis of Endometriosis"

_ijms, 2019, doi:10.3390/ijms20133372_

Round 1
Reviewer 1 Report
Comment on the manuscript:
“Role of B-cell translocation gene 1 in the3 pathogenesis of endometriosis"
The manuscript relates a study the purpose of which being to show if the downregulation of BTG1 (B-cell translocation gene 1), a translocation partner of c-myc plays a role in the endometriosis pathology. BTG-1 normally promotes apoptosis and negative-regulates cellular proliferation and adhesion between cells. For that, the authors studied several parameters on samples taken from patients with endometriosis. They also used transfected primarily cultured human endometrial stromal cells (HESCs).
This study performed in a wide sample constituted with 30 patients with endometriosis and 22 patients in control group (table 1) is interesting and useful. Nevertheless, I have some remarks about this manuscript.
Table 1. The number of patients in the endometriosis group is 30 women. Page 8, line 6, the authors indicate 54 women. There is a difference of 24 patients. Are they patients belonging to the group of excluded? It is not clear. If it is, indicate 24 exclusions.
Figure1B: give a scale bar on the pictures.
Figure 1C gives the staining intensity. Page 10, the authors indicate “the staining intensity of BTG1 was assessed on a scale of 0-3…”. What was the method used? a manual method? an automatic method? a semi-automatic method? If it's a manual method, how can you get a score of 2.5 or 1.5? This point needs to be clarified
Page 7, lines 24-30: Very honestly, the authors give some limitations of their study. In particular, they indicate the nature of the controls carried out on subjects suffering from a disease, which is explained by ethical reasons. I think this important clarification could also be given in the material and methods (page 8 "Patients and Tissue Samples").
Author Response
Point 1: Table 1. The number of patients in the endometriosis group is 30 women. Page 8, line 6, the authors indicate 54 women. There is a difference of 24 patients. Are they patients belonging to the group of excluded? It is not clear. If it is, indicate 24 exclusions.
Reply) As we already mentioned at Page 8, lines 18-22, 54 indicates a total number of patients who participated in this study. Thirty-two patients had peritoneal and/or ovarian endometriosis, whereas 22 patients were participated in the control group. Two of the 32 patients who had mild form of endometriosis were excluded in the analysis to preserve uniformity of disease group character. We added the sentence to Page 8, line 8.
Thirty-two and 22 patients participated in the endometriosis and control groups, respectively.
Point 2: Figure1B: give a scale bar on the pictures.
Reply) We inserted scale bars in the right lower corner of each image in the Figure 1B as the reviewer suggested.
Point 3: Figure 1C gives the staining intensity. Page 10, the authors indicate “the staining intensity of BTG1 was assessed on a scale of 0-3…”. What was the method used? a manual method? an automatic method? a semi-automatic method? If it's a manual method, how can you get a score of 2.5 or 1.5? This point needs to be clarified
Reply) The staining intensity of BTG1 was assessed by a gynecological pathologist, on a scale of 0-3 (0, negative staining; 1, weak; 2, moderate; and 3, strong. Each column with error bar in Figure 1C does not indicate an individual score, but the mean value of BTG1 staining intensity.
Point 4: Page 7, lines 24-30: Very honestly, the authors give some limitations of their study. In particular, they indicate the nature of the controls carried out on subjects suffering from a disease, which is explained by ethical reasons. I think this important clarification could also be given in the material and methods (page 8 "Patients and Tissue Samples").
Reply) We inserted the explanation for including patients with benign ovarian lesions as the control group in the Patients and Tissue Samples of the Materials and methods section.
These benign ovarian lesions were not associated with ovarian endometriosis or endometrial pathology. Since surgical resection of nonpathological ovarian or endometrial tissues from healthy patients raises ethical implications, the control group did not include disease-free subjects.
Reviewer 2 Report
The authors present a well designed work, which is definitively of interest for the readers.
I have some comments:
page 2, line 4: please submit a reference for the increased levels of estrogen
page 2, line 24: should read "affects"
page 7, line 24: to my opinion this is a pilot study - this should be mentioned as a limitation
page 7, line 40: could play an imortant
page 8, line 1: it is far to much to takl about new therapeutic options - first this role has to be proven by a validation study with a larger number of patients
page 8, line 12: there is no information about the pain scores of the patients. Was there any deep infiltration endometiosis detectable?
page 8, line 16: in which cycle phase were the endometrial specimen? what kind of lesions were take for analysis?
Author Response
Point 1: page 2, line 4: please submit a reference for the increased levels of estrogen
Reply) We added a reference (J Clin Endocrinol Metab. 2012 Nov; 97(11): 4228–4235) to Page 2, line 4.
Point 2: page 2, line 24: should read "affects"
Reply) We corrected the word as the reviewer suggested.
Point 3: page 7, line 24: to my opinion this is a pilot study - this should be mentioned as a limitation
Reply) We corrected the second sentence in the paragraph explaining the limitation.
There are a few limitations in this study. First, we conducted a pilot study using a relatively small number of the endometriosis subjects.
Point 4: page 7, line 40: could play an important
Reply) We corrected the phrase as the reviewer suggested.
Point 5: page 8, line 1: it is far too much to talk about new therapeutic options - first this role has to be proven by a validation study with a larger number of patients
Reply) We agree with the reviewer’s comment. We removed the last sentence as the reviewer suggested.
Point 6: page 8, line 12: there is no information about the pain scores of the patients. Was there any deep infiltration endometiosis detectable?
Reply) We assessed pain scales from patients with or without endometriosis using visual analogue scales (VAS) [Hum Reprod Update. 2015; 21: 136-52]. The mean values of VAS were 6.57 and 1.55 in the endometriosis and control groups, respectively. The differences in VAS between the groups were statistically significant (P<0.001). We included the values in the Results section and Table 1 of revised manuscript.
Since there are numerous researches documenting that deep infiltrating endometriosis (DIE) may have different pathophysiologic characteristics from those of ovarian/peritoneal endometriosis, we agree with the reviewer’s comment that it would be very interesting to see BTG1 expression pattern in DIE. Unfortunately, however, our study population did not include patients with DIE. It is definitely the point that warrants furtherinvestigation.
Point 7: page 8, line 16: in which cycle phase were the endometrial specimen? what kind of lesions were take for analysis?
Reply) As indicated in Table 2, the endometrial tissue samples were available in 29 of 30 patients with moderate-to-severe form of endometriosis and 18 of 20 control women.
In the endometriosis group, 14 and 15 endometrial tissue samples were those of proliferative and secretory phase, respectively. In the control group, 8 and 10 endometrial tissue samples were those of proliferative and secretory phase, respectively.
To remove the possible effect of other diseases, we included patients with nonpathological endometrium only and excluded cases with postmenopausal symptom, previous hormonal agent use, adenomyosis, endometrial polyp, hyperplasia, malignancy, infection, inflammation, or cardiovascular disease.